# The Synergistic Effect of Quince Fruit and Probiotics (*Lactobacillus* and *Bifidobacterium*) on Reducing Oxidative Stress and Inflammation at the Intestinal Level and Improving Athletic Performance during Endurance Exercise

**DOI:** 10.3390/nu15224764

**Published:** 2023-11-13

**Authors:** Karen Marlenne Herrera-Rocha, María Magdalena Manjarrez-Juanes, Mar Larrosa, Jorge Alberto Barrios-Payán, Nuria Elizabeth Rocha-Guzmán, Alejo Macías-Salas, José Alberto Gallegos-Infante, Saul Alberto Álvarez, Rubén Francisco González-Laredo, Martha Rocío Moreno-Jiménez

**Affiliations:** 1Research Group on Functional Foods and Nutraceuticals, Department of Chemical and Biochemical Engineering, TecNM/Instituto Tecnológico de Durango, Felipe Pescador 1830 Ote., Durango 34080, Mexico; karennerak91@gmail.com (K.M.H.-R.); magda1691@yahoo.com (M.M.M.-J.); nrocha@itdurango.edu.mx (N.E.R.-G.); agallegos@itdurango.edu.mx (J.A.G.-I.); 10040546@itdurango.edu.mx (S.A.Á.); rubenfgl@itdurango.edu.mx (R.F.G.-L.); 2Department of Nutrition and Food Science, Facultad de Farmacia, Universidad Complutense de Madrid, 28040 Madrid, Spain; mlarrosa@ucm.es; 3Laboratory of Experimental Pathology, National Institute of Medical Sciences and Nutrition Salvador Zubirán (INCMNSZ), Vasco de Quiroga #15, Tlalpan, Ciudad de México 14080, Mexico; jorge.barriosp@incmnsz.mx; 4Hospital Santiago Ramón y Cajal, Departamento de Patología, Instituto de Seguridad y Servicios Sociales de los Trabajadores del Estado, Durango 34079, Mexico; virjophd@gmail.com

**Keywords:** quince, probiotics, polyphenols, exercise endurance

## Abstract

Endurance exercise promotes damage at the intestinal level and generates a variety of symptoms related to oxidative stress processes, inflammatory processes, microbiota dysbiosis, and intestinal barrier damage. This study evaluated the effects of quince (*Cydonia oblonga* Mill.) and probiotics of the genera *Lactobacillus* and *Bifidobacterium* on intestinal protection and exercise endurance in an animal swimming model. Phytochemical characterization of the quince fruit demonstrated a total dietary fiber concentration of 0.820 ± 0.70 g/100 g and a fiber-bound phenolic content of 30,218 ± 104 µg/g in the freeze-dried fruit. UPLC-PDA-ESI-QqQ analyses identified a high content of polyphenol, mainly flavanols, hydroxycinnamic acids, hydroxybenzoic acids, flavonols, and, to a lesser extent, dihydrochalcones. The animal model of swimming was performed using C57BL/6 mice. The histological results determined that the consumption of the synbiotic generated intestinal protection and increased antioxidant (catalase and glutathione peroxidase enzymes) and anti-inflammatory (TNF-α and IL-6 and increasing IL-10) activities. An immunohistochemical analysis indicated mitochondrial biogenesis (Tom2) at the muscular level related to the increased swimming performance. These effects correlated mainly with the polyphenol content of the fruit and the effect of the probiotics. Therefore, this combination of quince and probiotics could be an alternative for the generation of a synbiotic product that improves exercise endurance and reduces the effects generated by the practice of high performance sports.

## 1. Introduction

Exercise promotes physiological health in athletes, which can potentially lower mortality and morbidity rates associated with chronic degenerative diseases such as cardiovascular diseases, diabetes, and cancer. Additionally, exercise can lead to muscular hypertrophy and increased endurance in sports [1,2]. Various forms of physical activity can be classified based on the degree of endurance, strength, speed, and mobility required. Some examples include swimming, marathons, and cycling [3]. Swimming is a sport in which athletes exceed their own endurance and increase the degree and time of exercise, which can generate physiological, biochemical, and molecular modifications that promote changes in the body, mainly at the gastrointestinal level, especially in the stomach, small intestine, and colon, and the appearance of oxidative stress and inflammation [4]. These modifications lead to gastrointestinal symptoms like nausea, vomiting, diarrhea, abdominal cramps, and rectal bleeding, which prompt most athletes to withdraw from competition [5].

The process by which these modifications are triggered in the body is splanchnic hypoperfusion. This process promotes changes in the systemic circulation in which blood flow is redistributed from the central area to the extremities, reducing the supply of oxygen and nutrients to vital organs such as the stomach, small intestine, and colon, affecting the intestinal barrier, specifically the mucosa and tight junction proteins, and promoting an inflammatory response, cellular infiltration, and ulceration [6]. During this process, an overproduction of reactive oxygen species (ROS) and reactive nitrogen species (RNS) leads to the transcription of nuclear factor kappa β (NF-κβ), which is involved in the transcriptional regulation of genes involved in the process of oxidative stress and inflammation, activating the expression of inflammatory cytokines and chemokines (IL-8, TNF-α and IL-6) and the decreased activity of anti-inflammatory interleukins (IL-10), and the elimination of oxidative stress by antioxidant enzymes such as catalase or glutathione peroxidase [7].

Athletes use pharmacological treatments, including non-steroidal anti-inflammatory drugs (NSAIDs), to reduce gastrointestinal symptoms; however, these drugs cause side-effects such as ulceration, altered renal function, and decreased cardiovascular function [8]. Therefore, athletes may consider consuming natural sources such as probiotics and prebiotics to enhance their beneficial effects on the host and alleviate symptoms. The combination of prebiotics and probiotics has shown superior synergistic effects in the body compared to their independent components. This has resulted in reduced gastrointestinal discomfort [9]. In particular, probiotics have been recognized for their ability to induce biological changes, including the regeneration of the microbiota, the enhancement of host immunity, and the reduction of LDL cholesterol levels [10]. The consumption of *Lactobacillus* and *Bifidobacterium* strains has been related to intestinal barrier regeneration, anti-inflammatory, and antioxidant effects [11]. Probiotics, such as *Lactobacillus casei* and *Lactobacillus paracasei*, have a significant impact on various diseases. Regenerating tight junction proteins and intestinal mucosa repairs the intestinal barrier, improves inflammatory processes by decreasing the expression of TNF-α, IL-6, and IL-1β, and eliminates oxidative stress, leading to a decrease in autoimmune, metabolic, and mental diseases [12]. Additionally, probiotics promote muscle health by increasing strength in muscles like the quadriceps and gastrocnemius [13]. Species such as *Bifidobacterium longum*, *B. breve*, and *B. bifidum* activate anti-inflammatory responses (IL-10), regulate intestinal epithelial function, synthesize antimicrobial compounds, regenerate tight junction proteins, and promote mitochondrial biogenesis in skeletal muscle through the AMPK-PGC-1α pathway, inducing muscle hypertrophy [14,15,16,17]. Additionally, research suggests that consuming fruits, which are prebiotic sources high in fiber and polyphenols, can enhance gastrointestinal symptoms by intervening in the intestinal tissue [18]. Quince (*Cydonia oblonga* Mill.) has traditionally been associated with the treatment of various pathologies. Several studies have reported on the health benefits of consuming different parts of the quince tree. According to Abliz et al. [19], the leaves were found to reduce or treat cardiovascular problems. The fruit has been utilized by the industry to obtain pectins for treating intestinal ulcers [20]. Additionally, extracts derived from the fruit have been shown to possess antioxidant and anti-inflammatory effects [21], as well as being anticarcinogenic [22] and antidepressant [23] in both in vivo and in vitro studies. The beneficial health effects of this fruit have been linked to its diverse phytochemical content, including its dietary fiber content which stimulates the production of short-chain fatty acids (SCFA), such as acetate, butyrate, and propionate, through microbial metabolism. These metabolites are significant bioproducts produced by the microbiota and contribute to gastrointestinal benefits by reducing pH levels and increasing microbial growth and diversity, promoting endocrine effects and communication of the gut–brain-microbiota axis [24]. Polyphenolic compounds, including hydroxycinnamic acids (such as chlorogenic acid and quinic acid), hydroxybenzoic acids (such as shikimic acid, ferulic acid, and gallic acid), flavonols (such as quercetin and kaempferol), and flavanols (such as catechin), are present. (Epi)-catechins are compounds that are associated with antioxidant effects due to their increased availability of hydroxyl (-OH) groups that promote the activation of pathways with anti-inflammatory effects, such as the nuclear factor-kappa β (NF-κβ) transcriptional pathway, which is involved in the regulation of genes that influence the process of oxidative stress and inflammation [25,26].

Therefore, the objective of this study was to determine whether there is a synergistic effect in the consumption of probiotic strains (*Lactobacillus* and *Bifidobacterium*) and quince (*Cydonia oblonga* Mill.) at the level of intestinal protection through the reduction of oxidative stress and proinflammatory processes, in addition to promoting an increase in athletic performance during swimming as an endurance exercise.

## 2. Materials and Methods

### 2.1. Chemical Reagents

Pepsin, pancreatin, α-amylase, amyloglucosidase, gallocatechin standards, catechin, epicatechin, procyanidin B2, quercetin, quercetin-3-O-glucoside, naringenin, naringin, luteolin, apigenin, acacetin, rutin, neohesperidin, taxifolin, floretin, kaempferol, kaempferol-3-O-glucoside, quinic acid, protocatechuic acid, 2,5-dihydroxybenzoic acid, 4-hydroxybenzoic acid, 2,4,6-trihydroxybenzaldehyde acid, syringic acid, chlorogenic acid, 4-O-caffeoylquinic acid, caffeic acid, coumaric acid, formic acid, shikimic acid, dextran sulfate sodium (DSS), and kerosene (327204) were purchased from Sigma Chemical (St. Louis, MO, USA). Acetonitrile, formic acid, ethyl acetate, ethanol, and xylol were purchased from J.T. Baker Inc., Phillipsburg, NJ, USA. Tris(hydroxymethyl)aminomethane, ethylenediaminetetraacetic acid, dithiothreitol, acrylamide, bisacrylamide, sodium dodecyl sulfate, ammonium persulfate, hydrogen peroxide, sodium azide, cumene hydroperoxide, lithium carbonate, and formaldehyde were purchased from ThermoFisher Scientific, Waltham, MA, USA. Tetramethylethylenediamine and Tween 20 were purchased from Bio-Rad, Waltham, MA, USA. Primary monoclonal antibodies TNF-α (SC-52746 Lot# L1517), IL-6 (SC-32296 Lot#G2617), IL-10 (SC-365062 Lot#D0716), Tom20 (SC-17764 Lot#), and secondary antibodies (m-IgGk BP-HRP SC-516102) were purchased from Santa Cruz Biotechnology Inc., Santa Cruz, CA, USA. Diaminobenzidine (ImmPACT^®^ DAB Substrate Kit, Peroxidase, SK-4105, Vector Laboratories, Newark, CA, USA), hematoxylin (2000937400, Biopack, Sydney, Australia), and eosin (2000937100, Biopack) were purchased from BIOPACK. Background Sniper (BS966M) and Super Pap Pen (PEN1111) were purchased from BioCare Medical, Makati, Philippines. Mouse-Rabbit ImmunoDetector Biotin Link (BSB 0.0001 L) was purchased from Bio SB, Goleta, CA, USA, Bioscience for the world.

### 2.2. Sample Processing

Quince (*Cydonia oblonga* Mill.) was collected in El Salto, Pueblo Nuevo, Durango, México (latitude 23.7794, longitude 150.362) during the September 2019 season. Samples were collected by MR Moreno-Jiménez and herbal verification was performed by Dr. Socorro González-Elizondo of the CIIDIR-IPN herbarium, Durango, México, accession number 58696. To obtain quince pulp, the fruit was sanitized according to NOM-251-SSA1-2009 and processed in an industrial pulper (Polinox, México City, México) at the Instituto Tecnológico de Durango, Mexico. The resulting pulp was stored at −20 °C and then lyophilized (0.045 mBar, −51 °C) (FreeZone 18 lyophilizer, LABCONCO, Kansas City, MO, USA) until further analysis.

### 2.3. Biological Material

*Lactobacillus casei* (ATCC 27045*), Lactobacillus paracasei* (ATCC 11582), *Bifidobacterium longum* (DSM-20219), *Bifidobacterium bifidum* (ATCC 29521), *Bifidobacterium breve* (ATCC 15700) strains were used. Male mice of the C57BL/6 strain, 10 weeks old (22.6 ± 2.5 g), were obtained from the Instituto de Neurología UNAM, Campus Juriquilla (Campus UNAM 3001, 76230 Juriquilla, Querétaro, México).

### 2.4. Determination of Dietary Fiber and Extraction of Bound Polyphenols from Quince (Cydonia oblonga Mill.)

The determination of dietary fiber was carried out according to the method used by Goñi et al. [27], with slight modifications. Samples were processed in triplicate at a concentration of 300 mg of quince, weighed, and resuspended in phosphate buffer (pH 7.5, 0.1 M). Then, 0.2 mL of pepsin was added and the samples were incubated at 40 °C for 1 h. Then, the pH was adjusted to 7.5 and 1 mL of pancreatin was added and the mixture incubated at 37 °C for 6 h. Then, 10 mL of trizma-maleate buffer (pH 6.9, 0.1 M) was added, and 1 mL of α-amylase was incubated for 16 h at 37 °C at constant stirring. The samples were centrifuged at 3000× *g* for 15 min and the supernatants were removed. The pellets were washed twice with distilled water and dried overnight at 105 °C, and the total insoluble fiber content was determined. To the supernatant fraction, 10 mL sodium acetate buffer (pH 4.75, 0.2 M) and 0.1 mL amyloglucosidase were added and incubated at 60 °C for 45 min. The resulting mixture was dialyzed with Spectrum™ Spectra/Por™ 4 RC Dialysis Membrane Tubing 12,000–14,000 Dalton MWCO from Sigma-Aldrich (St. Louis, MO, USA). with water exchanged every 24 h for 48 h at 37 °C. The weight of the collected samples was determined, corresponding to the soluble dietary fiber. Fiber-bound polyphenols were determined via liquid–liquid extraction (1:4 *v*/*v*) with ethyl acetate to obtain the sample, which was dried (Labconco Centrifugal Concentrator, Kansas City, MO, USA). Aliquots were stored for future analysis.

### 2.5. Analysis of Bound Polyphenols via UPLC-PAD/ESI-QqQ MS/MS

Samples were analyzed for the identification and quantification of polyphenol compounds according to the method used by Díaz-Rivas et al. [28]. Samples obtained from liquid–liquid extraction and dried via dialysis were resuspended in 200 µL of methanol and filtered through 0.45 µm PTFE filters for subsequent analysis on a UPLC system coupled to a Xevo TQ-S triple quadrupole tandem (Waters Corp., Milford, MA, USA).

Data were collected in the multiple reaction monitoring (MRM) mode. Data acquisition and processing were performed using MassLinx v. 4.1 Software (Waters Corp., USA). Chromatographic separations were carried out using a C18 Acquity UPLC BEH column (100 mm × 2.1 mm × 1.7 µm) (Waters Corp., USA) operating at 35 °C, with 7.5 mM water/formic acid (A) and acetonitrile (B) as mobile phases at 0.35 µL/min, using a sample volume of 2 µL. The gradient was applied at 3% B, held for 1.23 min, followed by 9% B at 3.82 min, followed by 16% B at 11.40 min, followed by 50% B at 13.24 min, followed by 3% at 15 min, returning to initial conditions (3% B). Negative ionization was used for MS, with ESI conditions as follows: capillary voltage, 2.5 kV; desolvation temperature, 300 °C; source temperature, 150 °C; desolvation and cone gas, 500 L/h and 151 L/h, respectively; collision gas, 0.13 mL/min. MRM transitions were determined via MS/MS spectra for the standards and the polyphenols present in the samples. Peak identification was based on the comparison of retention times and MRM transitions with the pure standards. Quantitative determinations of phenolic compounds were performed using calibration curves of the standards available from Sigma Chemical (St. Louis, MO, USA).

### 2.6. Isolation of Probiotics

*Lactobacillus casei* and *Lactobacillus paracasei* strains were resuspended in MRS medium. *Bifidobacterium longum* (DSM-20219), *Bifidobacterium bifidum*, and *Bifidobacterium breve* were resuspended in MRS supplemented with 0.05% cysteine. They were incubated at 37 °C under anaerobic conditions (CO_2_ 5%) for 24 h. Subsequently, a 10% inoculum of each bacterium was removed and resuspended in fresh MRS, and again incubated at 37 °C under anaerobic conditions (CO_2_ 5%) for another 24 h. All bacteria were administered together at a total cell density of 1 × 10^10^ cells/mL in rodents.

### 2.7. Swimming Exercise Model

C57BL/6 male mice (22.6 ± 2.5 g) were used. The animals were fed chow (LabDiet 5001 Rodent Diet) and water ad libitum during the acclimation week and the experimental period. They were kept in the vivarium under the following conditions: light–dark cycle (12:12 h), room temperature of 24 ± 1 °C, and relative humidity of 40–50%. Biosafety, experiments, and euthanasia of animals were carried out according to the animal care standards established by the Mexican Official Standard NOM-062-ZOO-1999 [29].

The administration of treatments lasted four weeks, after one week of acclimatization. The experimental groups were divided into four groups of 10 rodents each: control (−) (normodiet without damage inducer DSS 1.5%), control (+) (normodiet with damage inducer DSS 1.5%), probiotics (probiotics/damage inducer DSS 1.5%), and quince synbiotic (quince/probiotics/damage inducer DSS 1.5%). The probiotics were administered daily via gavage, in a volume of 0.200 mL and freeze-dried, and ground quince was administered in the cages at a dose of 2 g/animal/day. The total concentrations of probiotics and prebiotics are described in Table 1.

Body weight, food consumption, and water consumption were determined daily. During the third week, mice were trained to swim for 5 min (water temperature 30–32 °C) by administering doses of 1.5% dextran sulfate sodium (DSS) in the drinking water, allowing the animals to drink ad libitum at the same time as food and quince consumption. At the end of this week, the DSS was removed, and DSS-free water was again administered ad libitum. In the fourth week, the mice were subjected to 5 days of intensive swimming to exhaustion, with 5% of body weight added to the tail of each mouse. Swimming time was recorded daily on an individual basis in all groups to determine exercise capacity. On the fifth day of exercise, the mice swam until exhaustion, were euthanized using sodium pentobarbital as the anesthetic, and their heart punctured for subsequent collecting of organs, which were stored at −80 °C until analysis (Figure 1). The percentage of exercise capacity increase was determined by considering the initial and final exhaustion times.

### 2.8. Preparation of Organs for Histology and Immunohistochemistry

Segments of 1 cm of ileum and ascending colon were fixed in 10% formaldehyde. Gastrocnemius muscle was fixed in absolute ethanol. The tissues were then dehydrated (water, 70% ethanol, 90% ethanol, 96% ethanol, 100% ethanol, xylol) (Histokinette STP 129 Thermo Scientific), embedded in paraffin and sectioned at 4 µm (Leica RM 2145 microtome). The intestine and colon sections were placed on unloaded slides for histology (AmScope BS-50P-100S-22) and the muscles were placed on loaded slides for immunohistochemistry (Kling on HIER Slides, SFH1103A, BioCare Medical).

### 2.9. Obtaining the Protein Fraction from Small Intestine and Colon

Small intestine and colon were lysed with liquid nitrogen and resuspended in lysis buffer (TRIS 50 mM, EDTA 1 mM, DTT 1 mM, potassium acetate 100 mM, 1% protease inhibitor (Sigma-Aldrich), pH 7.8). They were then homogenized in Ultra-turrax (IKA^®^ T10) for four cycles of 30 s at 3000 rpm. The homogenates were centrifuged at 12,000 rpm for 10 min and the supernatants were collected. Total protein concentration was determined spectrophotometrically at 260 nm (Take3 Trio Micro-Volume Plate-Biotek).

### 2.10. Immunodetection of Inflammatory Markers

Protein homogenates from the small intestine and colon (100 µg/µL) were separated via electrophoresis on 15% acrylamide gels. The gels were transferred to nitrocellulose membranes (Thermo Scientific, 0.45 µm #10773485) using a transfer module (Trans-Bot Turbo, transfer system #1704150) for 7 min at 100 V. The membranes were blocked with nonfat milk (4% Svelty) for 1 h, then washed three times (5, 10, 15 min) with 1X TBS-Tween buffer (20 mM Tris/HCl, 100 mM NaCl and Tween 20, 0.2% *v*/*v*, pH 7.6). Incubation with the primary monoclonal antibodies TNF-α, IL-6, and IL-10 was performed at 16 °C for 8 h. After this time, washes were performed with TBS (1X) and secondary antibodies were incubated for 2 h at 16 °C. Additional washes were performed with 1X TBS (Tris/HCl 20 mM, NaCl 100 mM, pH 7.6), and a developer solution (Clarity Western ECL Substrate) was added according to the manufacturer’s specifications. Densitometric analysis was performed using a photodocumentation system (Gel Doc XR + Gel Documentation System, BioRad) and image analysis was performed using Image Lab software version 5.2.1.

### 2.11. Antioxidant Enzymes in the Small Intestine and Colon

#### 2.11.1. Catalase Determination

Catalase assay was carried out according to Weydert et al. [30], using 100 µg/µL of protein extract. First, a phosphate-buffered solution (50 mM, pH 7.4–7.8) was prepared using H_2_O_2_ 30 mM (30% *v*/*v*) (Abs 240 nm). Then, 4 mL of phosphate buffer was added to a 12 × 75 mm glass tube, equivalent in µL to 100 µg of tissue protein, and this mixture was divided in equal volumes into two quartz cells, to one of which 1 mL of PB was added and used as a blank, adjusted to zero in the spectrophotometer (HACH DR5000, UV-Bis). To the second cuvette, 1 mL of 30 mM H_2_O_2_ solution (30% *v*/*v*) was added. The samples were homogenized, immediately placed in the spectrophotometer and a programmed kinetic was started every 30 s in the 240 nm spectrum for 2 min. Concentrations between 25 and 50 µL of catalase standard (400 U/mL^−1^) were used for the reference curve. The results are expressed as mg mg^−1^ protein.

#### 2.11.2. Glutathione Peroxidase Assay

Glutathione peroxidase assay was performed [30] using a protein extract concentration of 100 µg/µL. Kinetics were programmed at a temperature of 37 °C at 340 nm, recording changes every 15 s for 5 min in UV/VS spectra (SynergyTM HT Multi-Detection Microplate Reader, Bio-Tek, Winooski, VT, USA). A 150 µL working solution (reduced glutathione 1.33 mM, glutathione reductase 66.5 U/mL^−1^), 10 µL NADPH (4 mM), and 100 µg/µL protein extract of the samples were added, homogenized, and incubated at room temperature for 5 min. Then, 20 µL of cumene hydroperoxide (1.5 mM) was added, the reaction was homogenized, and the enzyme kinetics was started. The concentration curve was plotted for glutathione peroxidase (200 U/mL) at concentrations of 0.02–0.10 U/mL GPx. Results were expressed as mg^−1^ protein equivalents.

### 2.12. Determination of Intestinal Inflammation via Histological Analysis

Slides containing intestine and colon were deparaffinized at a temperature of 65 °C for 30 min, then rehydrated (xylol 3 min, xylol 1 min, xylol/ethanol 100% 1 min, ethanol 100% 1 min, ethanol 96% 1 min, ethanol 90% 1 min, ethanol 70% 1 min, H_2_O 1 min). For hematoxylin and eosin staining, the slides were placed, in the following order, in hematoxylin, H_2_O 3 min, saturated Li_2_CO_3_ (25890-20-4) 1 min, acidified ethanol (0.5%) 5–10 s, H_2_O 1 min, ethanol 96% 1 min, eosin 15–30 s, ethanol 100% 1 min, ethanol 100%/xylol 1 min, xylol 1 min, xylol 1 min, and xylol 3 min. Finally, the samples were mounted (Entellan 107961, Merck Millipore) and observed under a microscope at 20X and then analyzed according to the level of histological damage as reported [31,32] (Table 2).

### 2.13. Mitochondrial Biogenesis via Immunohistochemical Analysis

Slides containing gastrocnemius muscle were deparaffinized at 60 °C for 20 min and rehydrated (xylol 3 min, xylol 1 min, xylol/ethanol 100% 1 min, ethanol 100% 1 min, ethanol 96% 1 min, ethanol 90% 1 min, ethanol 70% 1 min, H_2_O 1 min). Washes were performed with 0.05% PBS 1X-Tween 20 (pH 7.2–7.4) and the tissue section was delimited. Subsequently, 100 µL of H_2_O_2_ (3%) was added and the slides were kept under agitation at 50 rpm for 10 min. After this time, 100 µL of Background Sniper was added and the samples were vortexed for 10 min. Subsequently, 100 µL of Tom 20 primary antibody (1:100) was added and incubated overnight at room temperature with shaking. After washing with PBS/Tween 20, 100 µL of mouse-rabbit ImmunoDetector Biotin Link was added and incubated for 15 min with shaking. Finally, 50 µL of diaminobenzidine (1:100) was added and incubated for 10 min with shaking. The samples were contrasted via immersion in hematoxylin for 3 min, followed by rehydration in the following order: H_2_O 3 min, 96% ethanol 1 min, 100% ethanol 1 min, 100% ethanol/xylol 1 min, xylol 1 min, and xylol 3 min, and immediately mounted (Entellan 107961, Merck Millipore, Burlington, MA, USA). Slides were observed under a microscope at 20X and photographed for subsequent analysis using ImageJ Fiji software 1.51n. Results are expressed as mitochondrial staining intensity.

### 2.14. Statistical Analysis

All results were expressed as mean ± standard deviation. Data were analyzed via one-way ANOVA (*p* < 0.05). Principal component analysis (PCA) was used to determine correlations between variables. Statistical analyses were performed with IBM SPSS Statistics 22.0 software (IBM Corp., Endicott, NY, USA).

## 3. Results and Discussion

### 3.1. Dietary Fiber and Bound Polyphenols of Quince (Cydonia oblonga Mill.) as a Prebiotic Source of Synbiotics

It has been described that the combined consumption of prebiotics and probiotics promotes a symbiotic effect that enhances beneficial health effects. It is therefore relevant that the selection of matrices with dietary fiber content is a priority for the generation of synbiotic products. Dietary fiber is considered a potential prebiotic component due to its soluble and insoluble properties that, along with the bound polyphenols in its chemical structure, enhance this effect [33]. Fruits usually have a high percentage of dietary fiber; however, it is relevant to know the type of fiber present to determine the prebiotic effect at the intestinal level [34]. According to the results obtained, the total fiber content of quince was 0.820 ± 0.70 g/100 g of freeze-dried fruit, of which soluble fiber was 45.02 ± 0.80% and insoluble fiber was 54.97 ± 0.05%. Quince is a suitable prebiotic source due to its ability to modify the intestinal microbiota. It has been described that the consumption of insoluble fiber promotes a greater relative abundance of the phyla *Bacteroidetes*, *Euryarcheota*, and *Runinococcaceae* and the genera *Prevotella*, *Phascolarctobacterium*, *Coprococcus,* and *Leeia*. On the contrary, the consumption of soluble fiber promotes less bacterial diversity in the microbiota, favoring the growth of the phylum *Proteobacteria* and less of *Prevotellaceae* [12]. Likewise, dietary fiber, when fermented by the microbiota, produces SCFA, mainly acetate, propionate, and butyrate [35]. These compounds are relevant for their ability to regulate the expression of genes that act as inhibitors of histone deacetylases-regulated processes such as cell proliferation, apoptosis, and differentiation [36]. Additionally, they recognize the G protein-coupled receptors (GPCRs) expressed in different cells, among which propionate and butyrate activate the GPCR43 receptor present in colonocytes, adipose tissue, the nervous system, immune system cells, and the pancreas, allowing for the regulation of the immune system and energy metabolism [37,38]. In addition, SCFA participate in the maintenance of the intestinal barrier by promoting the production and secretion of intestinal mucus produced by bifidobacteria [39].

On the other hand, a total fiber-bound polyphenol content of 30,218 ± 104 µg/g of freeze-dried fruit was determined in quince. This type of polyphenols can be released in the gastrointestinal tract by digestive enzymes (such as α-glycosidase, α-amylase, lipase, pepsin, trypsin, and chymotrypsin) and microbiota that promote the cleavage of chemical bonds between the fiber and the bound polyphenols, making these compounds accessible for further metabolism, thereby favoring their prebiotic effect [40]. Specifically, it was established that quince has a higher abundance of flavonols (16,358.35 ± 745.24 µg/g), hydroxycinnamic acids (8866.15 ± 919.52 µg/g), hydroxybenzoic acids (4564.74 ± 125.09 µg/g), flavanols (324.55 ± 22.71 µg/g), and flavones (39.29 ± 10.81 µg/g), and in lower concentrations dihydrochalcones (33.08 ± 8.76 µg/g), flavanonols (4.03 ± 1.46 µg/g), flavanones (3.29 ± 1.56 µg/g), and xanthones (0.22 ± 0.00 µg/g) (Table 3). From this polyphenol’s identification, the presence of relevant compounds such as quercetin-3-O-glucoside, kaempferol-3-O-glucoside, quinic acid, caffeic acid, benzoic acid, 4-hydroxybenzoic acid, catechin, and epicatechin, have been described, with anti-inflammatory and antioxidant effects.

### 3.2. Histopathologic Evaluation of the Small Intestine and Colon after Consumption of Quince and Probiotic Strains in the Swimming Model

According to the results of the animal model, the consumption of quince and probiotic strains did not affect the water and food consumption of the animals over the course of the experiment. However, the body weight was modified over time. Control (−), control (+), and probiotic mice increased their body weight over the weeks, while the quince synbiotic mice maintained their body weight over the 4 weeks of the experiment (Table 4). It has been described that DSS promotes weight loss in animal models subjected to colitis induction. However, in this model, only doses of 1.5% were used as opposed to the dose reported to affect body weight and intestinal tissues, which is usually doses of more than 3% in in vivo models [41].

Resistance exercise, such as swimming, is a triggering factor of intestinal inflammation by promoting an increase in permeability due to the reduction of the intestinal mucosa [42]. When the mucosa is compromised, lesions are generated on the intestinal and colonic tissues and are classified into different levels of damage ranging from mild inflammation to the appearance of ulcers and cellular infiltration in the lesion [43,44]. The results of the histopathologic analysis showed a level of damage between 0 and 3 in both the small intestines and colons in the study groups (Figure 2). From the histological results, the classification of the effect on the intestine and colon is shown in Table 5. It was determined that the consumption of probiotics promoted an increase in mucosal inflammation in the small intestine and colon compared to the other groups. This indicates that, probably, the concentration of probiotics administered (1 × 10^10^ cells/mL) or the combination of strains (*L. casei*, *L. paracasei*, *B. longum*, *B. breve* and *B. bifidum*) together with the responses generated by swimming promoted an alteration in the homeostasis of the intestinal microbiota, promoting changes in the integrity of the intestinal epithelium mainly at the colonic level. In the small intestine the consumption of quince contributed to reducing the effects generated by the probiotics, presenting a minor damage. This group did not present atrophy, ulcerations or microabscesses, showing only level 1 damage in three individuals of the group with injury in the superficial epithelium. However, in the colon, a greater number of individuals with a higher degree of damage (levels 2 and 3) were observed, especially in individuals administered DSS, as this compound has been used to induce inflammation, mainly colitis and ulceration, in mouse models [45,46]. It has been described that the colon shows the most significant effect promoted by intensive exercise, with the appearance of lesions observed due to the redistribution of the blood flow to the muscles of the extremities [46]. Therefore, based on these results, it can be suggested that the consumption of quince contributes to intestinal protection and that this is related to the abundance of polyphenol compounds such as hydroxycinnamic acids, flavonols, and flavanols, since they have been shown to have effects on the regeneration and maintenance of the structure of the intestinal mucosa [47,48]. In addition to the above, these polyphenols also act as an energetic substrate for the microbiota-promoting bacteria to synthesize and convert them into glucuronidated, sulfated, and methylated secondary metabolites or SCFA that promote the protection and maintenance of the intestinal mucosa [49].

### 3.3. Antioxidant Effect on the Small Intestine and Colon after Consumption of Quince and Probiotic Strains during Swimming Exercise

Oxidative stress is one of the principal systemic damages induced by changes in the intestinal barrier caused by intensive endurance exercise [4]. Superoxide dismutase (SOD), catalase (Cat), and glutathione peroxidase (GPx) are antioxidant enzymes present in the body that are responsible for controlling the production of reactive oxygen and nitrogen species (ROS/RNS) by scavenging free radicals. SOD and Cat are responsible for catalyzing the reaction of harmful hydrogen peroxide (H_2_O_2_) into oxygen and water, which otherwise promotes oxidative damage that needs to be controlled [50]. GPx, in addition to catalyzing the reduction of H_2_O_2_ to oxygen and water, also reacts with lipid peroxidation products such as hydroperoxides (ROOH) produced by the oxidative stress caused in the process of splanchnic hypoperfusion at the intestinal level in sports practitioners [51,52]. The results indicated that the quince synbiotic promoted an increase in catalase (Cat) activation compared to the control and probiotic groups in the small intestine, while in the colon there were no statistical differences observed between quince, probiotics, and the control (−) (Figure 3a,b). However, the GPx activity of quince and probiotics treatments in the small intestine and colon did not generate an effect in comparison with the control (−). A smaller effect on glutathione peroxidase (GPx) activity was observed mainly in the probiotic and quince synbiotic group (Figure 3c,d). This result could be related to the Cat activity, because this enzyme generates the decrease in most free radicals (H_2_O_2_), catalyzing them to oxygen and water, and promoting a lower activation of GPx to carry out the radical scavenging process [50]. In contrast, the control (−) and control (+) groups showed higher GPx activity in the intestine and colon that can be related to the presence of oxidative stress caused by the swimming exercise and the induction of damage with DSS, promoting the activation of nuclear factor-kappa B (NF-κβ), which would be related to the regulation of genes that influence the process of oxidative stress and inflammation [51].

Polyphenol compounds are the main antioxidant agents in the body and their biotransformation by the microbiota can enhance this effect. Authors have indicated that probiotic genera such as *Lactobacillus* and *Bifidobacterium*, after using polyphenols as a prebiotic source, suppress the oxidative stress process by activating the nuclear factor erythroid 2-related factor 2 (Nrf2/Keap1) pathway, controlling ROS gene transcription, and activating enzymes such as Cat and GPx [52,53].

Principal component analysis (PCA) indicated that in the small intestine, compounds such as gallic acid, ferulic acid, taxifolin, and mangiferin were associated with catalase activity (Figure 4). Authors have noted the intervention of these compounds in catalase production in vivo and in vitro models [54,55,56,57]. While no direct correlation of GPx with phenolic compounds was observed, the experimental effect on this enzyme may be related to the effect of postbiotics generated by the microbiota such as SCFA or the bioconversion of polyphenols with relevance to antioxidant processes in the body [58,59]. At the colonic level, 2,5-hydroxybenzoic acid, 4-hydroxybenzoic acid, shikimic acid, and vanillic acid were associated with catalase activity, while quinic acid, chlorogenic acid, protocatechuic acid, quercetin, quercetin-3-O-glucoside, and kaempferol-3-O-glucoside were associated with glutathione peroxidase activity (Figure 4). It has been previously described that these polyphenols are related to the activation of the GPx enzyme, thus maintaining the integrity of the intestinal barrier in in vivo models [60,61].

### 3.4. The Anti-Inflammatory Effect in the Small Intestine and Colon after Consumption of Quince Synbiotics and Probiotic Strains during Swimming Exercise

The inflammatory response develops when the activation of the transcription factor NF-kβ/IκB is promoted in response to tissue damage, promoting an increase in the levels of nitric oxide (NO), chemokines such as IL-8, cytokines, and proinflammatory interleukins such as TNF-α and IL-6 [22]. Swimming, being an intensive endurance exercise and causing the presence of an ischemic process via splanchnic hypoperfusion, may initiate the activation of inflammatory pathways that cause damage to vital organs, especially organs located in the central area of the body such as the small intestine and colon [51]. The results showed an anti-inflammatory effect in both organs by decreasing the accumulation of pro-inflammatory markers TNF-α and IL-6 and increasing the levels of IL-10 accumulation, with the greatest effect generated by the consumption of quince synbiotic (Figure 5). Correlation analysis showed that polyphenols such as 2,5-hydroxybenzoic acid, 4-hydroxybenzoic acid, shikimic acid, chlorogenic acid, benzoic acid, quinic acid, vanillic acid, caffeic acid, floridzin, rutin, quercetin, catechin, and epicatechin were involved in reducing TNF-α activity in the small intestine. In the colon, compounds such as shikimic acid, quercetin-3-O-glucoside, kaempferol-3-O-glucoside, ferulic acid, and chlorogenic acid were associated with the reduction of TNF-α (Figure 6). In other inflammatory models, hydroxybenzoic acids such as shikimic, vanillic, benzoic, 2,5-hydroxybenzoic, and 4-hydroxybenzoic acids have been associated with the attenuation of inflammatory responses and intestinal mucosal damage in DSS-induced colitis [62]. Likewise, hydroxycinnamic acids and their derivatives, such as chlorogenic acid, quinic acid, caffeic acid, and ferulic acid, have been shown to reduce inflammation at the intestinal level by interfering with TNF-α activity [61], while flavonoids such as rutin, catechin, epicatechin, quercetin, and their glycosidic derivatives, as well as dihydrochalcone floridzin, reduce the levels of TNF-α biomarker accumulation [63].

IL-6 is a marker of an inflammatory nature, and the consumption of probiotics and quince favored the decrease of this biomarker (Figure 5c,d). Unlike TNF-α, some other polyphenols were more specifically related. For example, in the small intestine, the relationship between naringenin and IL-6 was observed, and in the colon, polyphenols such as quinic acid, 5-hydroxybenzoic acid, and quercetin were correlated with the decreased activity of this biomarker (Figure 6). It has been described that the intervention of polyphenols has been related to promoting an anti-inflammatory effect provided directly to IL-6, since they activate the Nrf2/Keap1 pathway, which interferes in the decrease of this interleukin [64,65]. Similarly, compounds such as quinic acid, 5-hydroxybenzoic acid, and quercetin are associated with anti-inflammatory effects [62,63]. However, naringenin is a flavanone that has been shown to interfere with IL-6, promoting anti-inflammatory effects specifically through the activation of the transcription factor NF-kβ/IκB [66].

On the other hand, IL-10 is considered a potent anti-inflammatory marker and its action decreases gene expression and the synthesis of inflammatory markers. The results obtained show an anti-inflammatory effect with the consumption of quince and probiotics (Figure 5e,f). Despite their proximity in the principal component analysis, the anti-inflammatory effect was negatively correlated with the polyphenols present in quince (Figure 6). This could be explained by the microbiota metabolism of the intestine after using the fruit as a prebiotic source. The synthesis of primary metabolites (SCFA) and secondary metabolites, which are glucuronidated, sulfated, and methylated polyphenols, by the microbiota bacteria, could be involved in the activation of signaling pathways related to IL-10 synthesis and thus show an anti-inflammatory effect in the small intestine and colon, as observed by other researchers [67,68].

### 3.5. Increased Exercise Endurance and Mitochondrial Biogenesis in Muscles with the Consumption of Quince Synbiotics

The consumption of synbiotics by athletes and their relationship to increased endurance is one of the trends that has not been fully explored. It has been described that muscle is a tissue that adapts rapidly and continuously to conditions of inactivity or exercise. In this sense, mitochondrial biogenesis is a crucial step for the increase of muscle mass and endurance. This process involves the transcription of several proteins, including Tom20, a protein that is part of the outer membrane of the mitochondria, promoting greater exercise endurance [69,70].

The percentage of exercise capacity was determined by considering the initial and final exhaustion times of mice (Table 6). The results showed that the consumption of quince synbiotic led to higher endurance in mice subjected to swimming, with an increase in sports endurance of 108.31 ± 19.04%, followed by the consumption of probiotics at 99.79 ± 12.96%, and the positive control and negative control groups up to 71.48 ± 14.20% and 62.72 ± 10.71%, respectively. These results are correlated with the immunohistochemical analysis performed, in which a greater accumulation of mitochondria at the muscle level was determined in relation to the endurance in high performance sports in the group that consumed quince synbiotic (Figure 7a–d), as was the greater accumulation of Tom 20 (Figure 7e). In this sense, the effect can be related to the consumption of the components of the synbiotic (fiber, polyphenols, and probiotics). Quince has a high abundance of chlorogenic acid, gallic acid, and rutin, that could generate a greater mitochondrial biogenesis not only in the expression of Tom20 but also through the enzymatic activation of cytochrome synthase, cytochrome C oxidase, and the activation of mitochondrial agents [71,72]. Another compound abundant in quince is epicatechin, which was shown to stimulate mitochondrial volume, rib density, and protein markers in the skeletal muscle of mice [73]. In addition, authors have explained that soluble fiber consumption promotes greater exercise endurance and muscle gain through microbiota intervention [74].

On the other hand, the *Lactobacillus* and *Bifidobacterium* strains administered as the probiotic part of the synbiotic could contribute to the effect of mitochondrial biogenesis. These strains, when consuming prebiotic sources such as polyphenols and fermenting soluble quince fiber, synthesize butyrate that can be converted by other bacteria to secondary metabolites (sulfated polyphenols, glucuronidated, and methylated polyphenols). The latter may result in the indirect production of metabolites involved in muscle mass and strength gains [75], as well as promoting an improvement in physical activity and exercise endurance, and supporting the post-exercise recovery process [76,77,78].

## 4. Conclusions

The negative effects generated by the performance of endurance exercise were reduced by the synergy of the consumption of probiotic strains and quince. The polyphenolic composition and fiber content of quince, together with the effects of probiotics, produced a positive effect in the maintenance of the epithelial barrier and in the regulation of oxidative stress. They promoted the activation of the antioxidant enzymes catalase and glutathione peroxidase, combined with an anti-inflammatory effect by reducing the accumulation of the biomarkers TNF-α and IL-6 and increasing the levels of IL-10. Such accumulation was generated mainly due to the polyphenolic compounds of the group of hydroxybenzoic acids, hydroxycinnamic acids, flavonols, flavanols, flavanones, and dihydrochalcones present in quince. Besides the muscular level, the quince synbiotic has also improved the process of mitochondrial biogenesis and increased exercise endurance.

## Figures and Tables

**Figure 1 nutrients-15-04764-f001:**
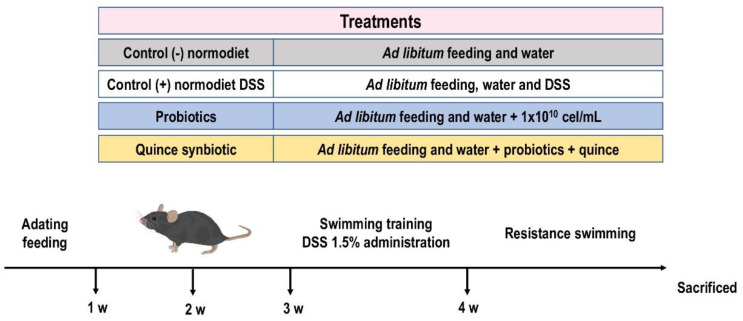
Experimental animal model of endurance swimming with the administration of synbiotic (quince and probiotic strains).

**Figure 2 nutrients-15-04764-f002:**
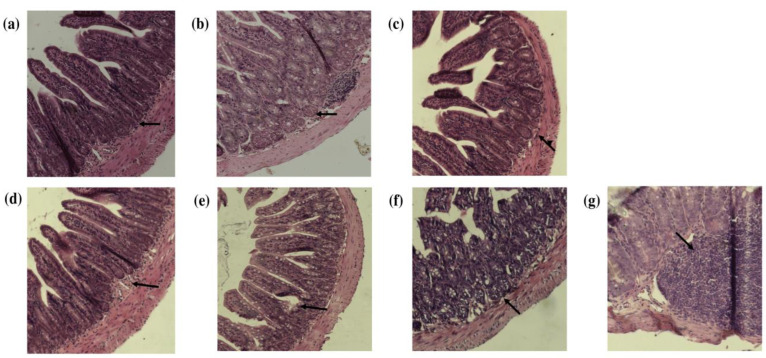
Small intestine and colon sections from C57BL/6 mice stained with hematoxylin/eosin and observed under a light microscope at 20X. Visual level of tissue damage. (**a**) Small intestine tissue with normal epithelium (level 0). (**b**) Small intestine tissue with inflammation of the epithelium (level 1). (**c**) Small intestine tissue with inflammation in the lamina propria and epithelium (level 2). (**d**) Colon tissue with normal epithelium (level 0). (**e**) Colon tissue with an inflamed epithelium (level 1). (**f**) Colon tissue with inflammation in the lamina propria and epithelium (level 2). (**g**) Colon tissue with leukocyte infiltration and inflammation in the lamina propria and epithelium (level 3).

**Figure 3 nutrients-15-04764-f003:**
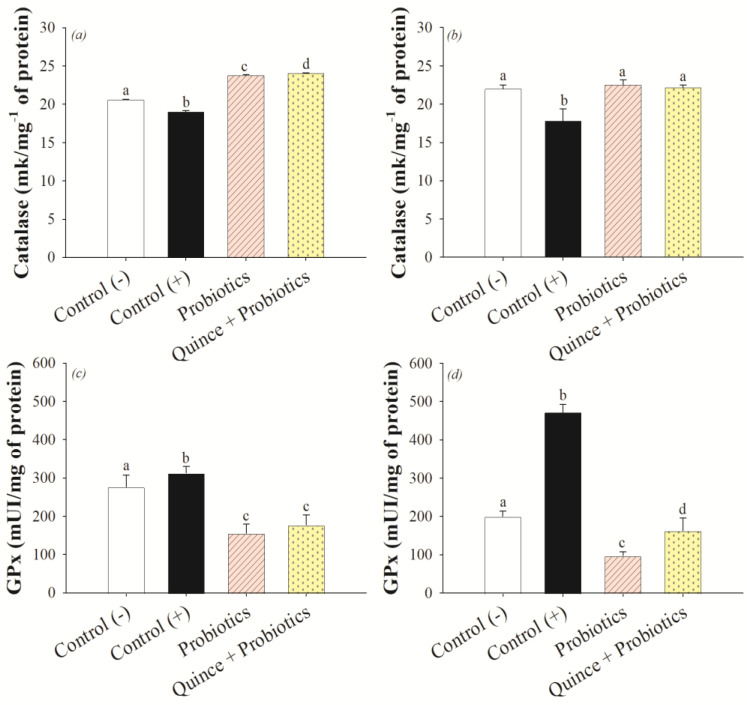
Antioxidant activity of catalase (Cat) and glutathione peroxidase (GPx) enzymes in small intestine and colon: (**a**) Cat activity of small intestine, (**b**) Cat activity of colon, (**c**) GPx of small intestine, and (**d**) GPx of colon. Different letters indicate statistical significance (ANOVA, Tukey *p* < 0.001).

**Figure 4 nutrients-15-04764-f004:**
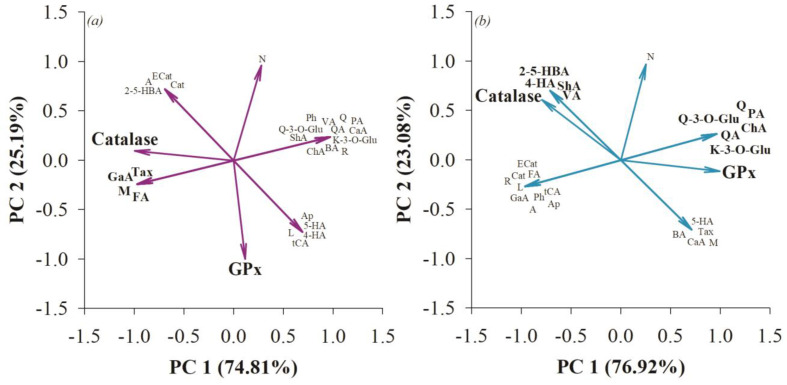
Principal component analysis (PCA) to determine the association of fiber-bound polyphenols from quince (*Cydonia oblonga* Mill.) with antioxidant processes in the small intestine (**a**) and colon (**b**).

**Figure 5 nutrients-15-04764-f005:**
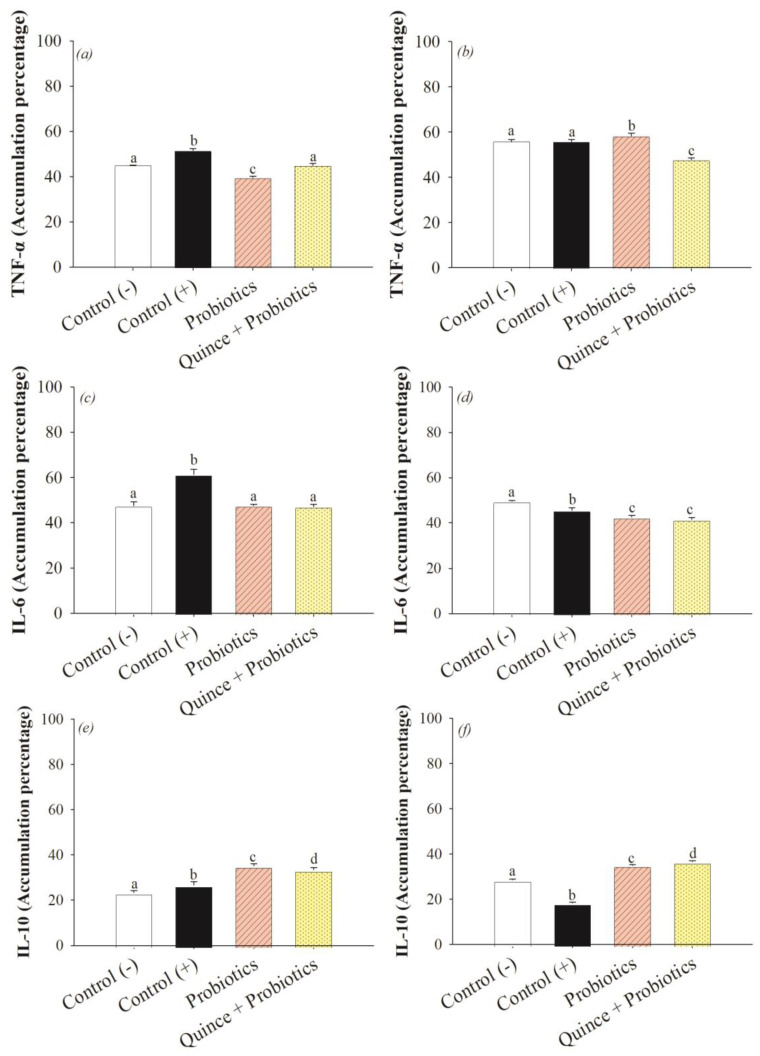
Accumulation levels of inflammation biomarkers at the intestinal and colonic levels. Different letters indicate statistical significance (ANOVA, Tukey *p* < 0.001). (**a**) TNF-α small intestine. (**b**) TNF-α colon. (**c**) IL-1 small intestine. (**d**) IL-6 colon. (**e**) IL-10 small intestine. (**f**) IL1-0 colon.

**Figure 6 nutrients-15-04764-f006:**
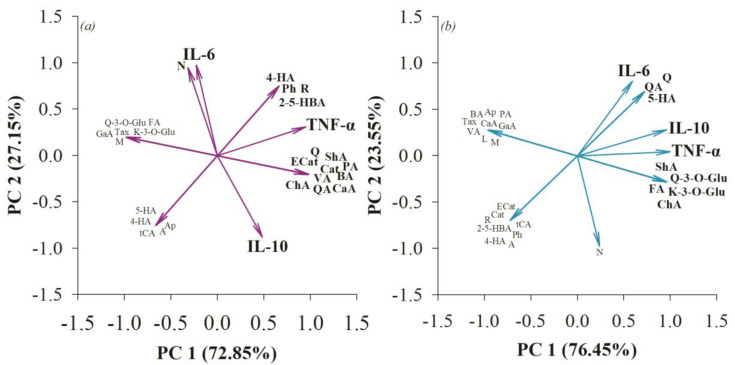
Principal component analysis (PCA) to determine the correlation of polyphenols linked to dietary fiber from quince (*Cydonia oblonga* Mill.) to anti-inflammatory processes in the small intestine (**a**) and colon (**b**).

**Figure 7 nutrients-15-04764-f007:**
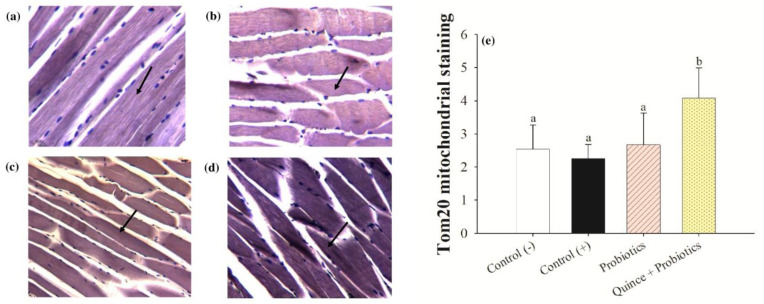
Gastrocnemius muscle sections observed via optical microscopy at 20X. (**a**) Control (−) (normodiet without DSS 1.5%). (**b**) Control (+) (normodiet/DSS 1.5%). (**c**) Probiotics (probiotics/DSS 1.5%). (**d**) Quince synbiotic (quince/probiotics/DSS 1.5%). (**e**) Tom 20 mitochondrial staining representing muscle improvement. Arrows indicate stained mitochondria. Different letters indicate statistical significance (ANOVA, Tukey *p* < 0.001).

**Table 1 nutrients-15-04764-t001:** Composition of Quince (*Cydonia oblonga* Mill.) and probiotic bacteria.

Prebiotics	Total Abundance
*Total of dietary fiber*	0.820 ± 0.70 g/100 g
Soluble fiber	45.02 ± 0.80%
Insoluble fiber	54.97 ± 0.05%
*Total of bound polyphenols*	30,218 ± 104 µg/g
**Probiotics**	**Total Dose of Bacteria**
*L. casei*	2 × 10^9^ cel/mL
*L. paracasei*	2 × 10^9^ cel/mL
*B. longum*	2 × 10^9^ cel/mL
*B. breve*	2 × 10^9^ cel/mL
*B. bifidum*	2 × 10^9^ cel/mL
*Total dose of probiotics*	1 × 10^10^ cel/mL

**Table 2 nutrients-15-04764-t002:** Classification of intestinal and colonic histological damage.

Level	Type of Injury and/or Damage
0	Normal, no damage
1	Inflammation in the superficial epithelium or mild cellular alterations such as atrophy
2	Inflammation in the lamina propria and epithelium
3	Inflammation in the lamina propria, epithelium, and muscle phase, leukocyte infiltration
4	Inflammation in the lamina propria, epithelium, muscle phase, leukocyte infiltration, and presence of ulcers
5	Inflammation in the lamina propria, epithelium, muscle phase, ulcers, and presence of microabsessions

**Table 3 nutrients-15-04764-t003:** Abundance of the bound polyphenols identified in Quince (*Cydonia oblonga* Mill.) via LC-PDA-ESI-QqQ.

No.	Compound	Acronym	Retention Time (min)	Molecular Weight	Main Transition (*m*/*z*)	λ Max (nm)	Abundance
	**Hydroxybenzoic acids**						
1	Gallic acid	GaA	1.07	169	79 > 125	270	4.17 ± 0.72
2	Vanillic acid	VA	3.71	167	123 >152	270	46.37 ± 4.63
3	Shikimic acid	ShA	0.57	173	93 > 111	270	233.77 ± 10.35
4	Proteocatechuic acid	PA	2.11	153	109 > 91	270	169.54 ± 5.95
5	2,5-dihydroxybenzoic acid	2-5-DHA	2.83	153	109 > 81	270	15.66 ± 1.46
6	4-hydroxybenzoic acid	4-HA	3.13	137	93 > 65	270	651.49 ± 17.32
7	5-hydroxybenzoic acid	5-HA	6.15	153	136 > 93	270	30.90 ± 0.59
8	Benzoic acid	BA	6.29	121	77 > 92	270	2954.54 ± 19.30
	**Hydroxycinnamic acids**						
9	Quinic acid	QA	0.56	191	93 > 85	320	759.23 ± 6.38
10	Caffeic acid	CaA	3.85	179	135 > 89	320	8370.80 ± 166.66
11	Chlorogenic acid	ChA	3.42	353	191 > 85	320	14.60 ± 0.42
12	Ferulic acid	FA	5.57	193	178 > 134	320	79.01 ± 4.11
13	*t*-cinnamic acid	tCA	8.44	148	148 > 149	320	481.59 ± 8.48
	**Flavonols**						
14	Quercetin	Q	8.28	480	479 > 303	360	175.94 ± 10.08
15	Quercetin-3-O-glucuronide	Q-3-O-Glu	5.94	480	479 > 303	360	9894.17 ± 98.47
16	Kaempferol-3-O-glucuronide	K-3-O-Glu	6.62	447	255 > 284	360	5859.11 ± 137.59
17	Rutin	R	5.84	610	609 > 300	360	553.92 ± 20.64
	**Flavanols**						
18	Catechin	Cat	3.16	289	245 > 123	280	47.70 ± 7.20
19	Epicatechin	ECat	4.38	298	245 > 123	280	276.84 ± 10.23
	Flavanones						
20	Naringenin	N	9.06	271	151 > 119	280	3.89 ± 0.90
	Flavones						
21	Luteolin	L	8.22	286	285 > 133	280	65.85 ± 5.52
22	Acacetin	A	11.37	269	148 > 117	320	25.88 ± 4.06
23	Apigenin	Ap	9.07	270	268 > 116	320	13.41 ± 3.90
	**Dihydrochalcones**						
24	Phloretin	P	9.21	273	123 > 167	280	0.00 ± 0.00
25	Phloridzin	Ph	7.35	436	435 > 167	280	33.08 ± 6.21
	Flavanonols						
26	Taxifolin	Tax	5.95	125	303 > 285	290	4.03 ± 0.84
	Xanthonoid						
27	Mangiferin	M	4.17	422	422 > 303	370	0.22 ± 0.00

The abundance of each polyphenol is expressed in µg/g of lyophilized fruit. All results were expressed as the mean ± standard deviation.

**Table 4 nutrients-15-04764-t004:** Body weight chance (g/animal/day).

Treatments	Week 1	Week 2	Week 3	Week 4
Control (−)	0.94 ± 0.13 ^b^	0.78 ± 0.18 ^c^	1.19 ± 0.58 ^a^	1.42 ± 0.48 ^a^
Control (+)	0.53 ± 0.12 ^a^	0.59 ± 0.10 ^c^	0.21 ± 0.05 ^c^	0.80 ± 0.13 ^d^
Probiotics	0.37 ± 0.08 ^a^	0.41 ± 0.08 ^c^	0.60 ± 0.08 ^d^	0.54 ± 0.05 ^c^
Quince synbiotic	0.56 ± 0.11 ^a^	0.52 ± 0.15 ^c^	0.51 ± 0.11 ^d^	0.59 ± 0.12 ^c^

All results are expressed as the mean ± standard deviation. Different letters indicate statistical significance (ANOVA, Tukey *p* < 0.001).

**Table 5 nutrients-15-04764-t005:** Scale of levels of histological damage in the small intestine and colon.

Small Intestine				
Level Damage	0	1	2	3
Control (−)	8 (10)	2 (10)	0 (10)	0 (10)
Control (+)	7 (10)	3 (10)	0 (10)	0 (10)
Probiotics	3 (10)	6 (10)	1 (10)	0 (10)
Quince synbiotic	7 (10)	3 (10)	0 (10)	0 (10)
**Colon**				
Level damage	0	1	2	3
Control (−)	5 (10)	3 (10)	2 (10)	0 (10)
Control (+)	1 (10)	4 (10)	5 (10)	0 (10)
Probiotics	3 (10)	5 (10)	1 (10)	1 (10)
Quince synbiotic	3 (10)	3 (10)	3 (10)	1 (10)

**Table 6 nutrients-15-04764-t006:** Swimming time during intensive exercise (min/day).

Groups	Day 1	Day 2	Day 3	Day 4	Day 5
Control (−)	9.47 ± 1.18	7.14 ± 0.67	5.43 ± 0.46	5.01 ± 0.29	2.20 ± 0.35
Control (+)	5.76 ± 0.69	4.05 ± 0.49	3.93 ± 0.34	3.44 ± 0.39	3.78 ± 0.60
Probiotics	9.48 ± 0.51	8.76 ± 0.68	8.32 ± 0.72	9.18 ± 1.14	5.66 ± 0.66
Quince synbiotic	9.07 ± 1.22	4.51 ± 0.28	7.88 ± 0.92	8.44 ± 1.12	2.52 ± 0.35

The results are expressed in mean ± standard deviation.

## Data Availability

The data used to support the findings of this study are available upon request to the corresponding author, mrmoreno@itdurango.edu.mx.

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
