# Peer review of "The Synergistic Effect of Quince Fruit and Probiotics (Lactobacillus and Bifidobacterium) on Reducing Oxidative Stress and Inflammation at the Intestinal Level and Improving Athletic Performance during Endurance Exercise"

_nutrients, 2023, doi:10.3390/nu15224764_

Round 1
Reviewer 1 Report
Comments and Suggestions for Authors
Currently, the relationship between microbiota and human physical activity has been proven. The human gut microbiota varies across different sports disciplines, and certain sports may promote the development of specific microbial ecosystems. The microbiome of «elite athletes» has its own characteristics (composition of species, properties of strains). Regular (long-term, from several weeks to several months) intake of probiotic compositions (certain strains of probiotic bacteria with specific properties) correlates with improved athletic performance and faster recovery of the physical condition of athletes.
This research shows synergistic effect in the consumption of probiotic strains and quince at the level of intestinal protection by reducing oxidative stress and proinflammatory processes.
The article is well-organized and contain all of the main components: Introduction, Materials and methods, Results and discussion, Conclusions. The sections are well-developed. The references used 77 sources, of which 70% sources for 2019-2023 years.
The methodology is clearly explained and the theory connects to the data. The text is clear and readable. The experimental findings justify the conclusions drawn. The authors showed that the polyphenolic composition and fiber content of quince, together with the effects of probiotics, produced an effect in the maintenance of the epithelial barrier and in the regulation of oxidative stress and reduce the level of TNF-α and IL-6. The manuscript provides an interesting insight and professional tone.
Author Response
We are thankful for the time and assistance; every issue and observations were attended carefully and highlighted properly in the revised manuscript. On behalf of all authors.
Dra. Martha Rocío Moreno-Jiménez
Thanks, we really appreciate attention and feedback from the reviewers
Reviewer 2 Report
Comments and Suggestions for Authors
1. Line 203: “…a total cell density of 1x 1010 CFU/mL in rodent.” Could you clarify the volume/amount instead of density CFU administered to each rodent? It is great that you made a table showing the ratio of each probiotics used. I believe it will read better if you refer to Table 1 in the methodology section.
2. Line 226: Could you specify the duration of time they swim on each occasion? Does the swimming time vary among groups?
3. Line 356: please specify what are the “digestive enzymes” responsible for the cleavage of the polyphenols.
4. Line 366-367: I didn’t find the anti-inflammatory and anti-oxidant effects in Table 3.
5. Line 384-386: what is the purpose/significance of mentioning 3% DDS treatment?
6. Figure 2: Could you specify which treatment group these H&E images belong to? Typically, it is advisable to include at least one representative image from each treatment group for each type of tissue.
7. Table 5: what do the values 0, 1, 2, 3 mean? Is it in weeks after probiotics treatment?
8. It seems that the probiotics and quince treatment group are more effective in promoting intestinal Catalase over GPx. Could you confirm this?
9. Do the probiotics and quince treatment groups also show an elevated level of polyphenol?
10. The use of PCA here is an innovative approach for analyzing the relationship between Cat/GPx and polyphenols.
11. Figure 4: a brief description of each abbreviation used in the figure is necessary.
12. Line 517: I think maybe you are referring to Figure 6 instead of 5?
13. Line 527-528: It appears that anti-inflammatory cytokines and quince may be negatively correlated based on your PCA.
Author Response
We are thankful for the time and assistance; every issue and observations were attended carefully and highlighted properly in the revised manuscript. On behalf of all authors.
Dra. Martha Rocío Moreno-Jiménez
- Line 203: "...a total cell density of 1x 1010 CFU/mL in rodent." Could you clarify the volume/amount instead of density CFU administered to each rodent? It is great that you made a table showing the ratio of each probiotics used. I believe it will read better if you refer to Table 1 in the methodology section.
Response: Dear Reviewer, we appreciate the observation. The volume administered was 0.200 mL, the concentration corresponds to a dose of 1x1010 cells/mL, which would be equivalent to the suggested dose to be consumed daily in humans. The data was already included in the text (line 216) as well as in reference to the doses presented in Table 1.
- Line 226: Could you specify the duration of time they swim on each occasion? Does the swimming time vary among groups?
Response: Dear reviewer, thank you for your comment, the swimming time has been integrated in Table 6 and the wording of this result has been modified because it had errors that did not allow to understand the results obtained (line 569-567)
My sincere apologies
3 Line 356: please specify what are the "digestive enzymes" responsible for the cleavage of the polyphenols
Response: This information was included in the manuscript (lines 357-358) thank you.
- Line 366-367: I didn't find the anti-inflammatory and anti-oxidant effects in Table 3.
Response: Thank you for the information, the relocation of table 3 in the paragraph (line 364) was made, which is really where it should be placed.
- Line 384-386: what is the purpose/significance of mentioning 3% DDS treatment?
Response: We consider it relevant to include the information in which it is stated that in vivo studies performed by other researchers normally use a dose equal to or greater than 3% DDS. However, a preliminary study performed by our research group indicated that 1.5% DSS was sufficient to generate inflammation in the colon and intestine, observing the damage at the histological level and determining of inflammatory markers.
- Figure 2: Could you specify which treatment group these H&E images belong to? Typically, it is advisable to include at least one representative image from each treatment group for each type of tissue.
Response: We appreciate the observation that Figure 2 was included in the manuscript to make a representation of the levels of damage that can be generated in the colon and intestine tissue as described by Bertevello & Erben, which served to make comparisons of the tissues from different groups and obtain the results described in Table 5.
- Table 5: what do the values 0, 1, 2, 3 mean? Is it in weeks after probiotics treatment?
Response: Thank you for your useful observation. The numbers indicate the level of histological damage that occurred in colon and intestine. This information was already included in Table 5 of the manuscript.
- It seems that the probiotics and quince treatment group are more effective in promoting intestinal Catalase over GPx. Could you confirm this?
Response: We appreciate your comments, the wording of this information was somewhat confusing, so additional and precise information was integrated (lines 450-452).
- Do the probiotics and quince treatment groups also show an elevated level of polyphenol?
Response: We appreciate the comment, this information was not included. However, a metabolomic study of serum, urine and feces could determine polyphenols as biomarkers of intake and establish the levels of polyphenols present in each group. We will take this into account when continue with research derived from this study.
- The use of PCA here is an innovative approach for analyzing the relationship between Cat/GPx and Polyphenols.
Response: Thanks, we appreciate the comment.
- Figure 4: a brief description of each abbreviation used in the figure is necessary
Response: We appreciate the suggestion, however Figure 4 has the acronyms of the polyphenolic compounds previously described in Table 3, so we do not consider it necessary to repeat them again.
- Line 517: I think maybe you are referring to Figure 6 instead of 5?
Response: Correction made, thank you
- Line 527-528: It appears that anti-inflammatory cytokines and quince may be negatively correlated based on your PCA.
Response: Thank you for your comment, the observation was corrected in the manuscript (lines 530-532)
